# Prevalence of musculoskeletal injuries among university undergraduates following Sri Lankan traditional dancing

**Geethika Chathurani, Yasantha B. Dassanayake, Sanduni N. Fernando, Lahiru S. Gunarathna, Lakshani K. Gunarathne, Nadheera C. Chandrasekara, Dilhari Senarath, Surangika I. Wadugodapitiya**⬤*

Department of Physiotherapy, Faculty of Allied Health Sciences, University of Peradeniya, Peradeniya, Sri Lanka

* surangikaw@ahs.pdn.ac.lk

**Data Availability Statement:** All relevant data are within the manuscript and its Supporting Information files.

## Abstract

Dancing is a demanding form of art that consists of a purposefully selected series of human movements presented in a rhythmic way. However, Dancers represent a medically under-served occupational group who are at high risk for work-related musculoskeletal disorders. Injury prevention among dancers has become challenging due to the dearth of research in the field. Therefore, a cross-sectional survey to determine the prevalence of musculoskeletal injuries among Sri Lankan traditional dancers is vital of need. The main objective of this research was to evaluate the prevalence of common musculoskeletal injuries among university undergraduates who follow Sri Lankan traditional dancing. This study was a cross-sectional descriptive study that included undergraduate students from four local universities. Stratified sampling method was used to select 293 participants and an online questionnaire was used to collect data. Among the three types of traditional dancing styles, many were following Kandyan dancing: 45.1%. Out of the study sample, 190 dancers (64.84%) reported injuries with males indicating the highest rate of injuries (36.87%). The most common injury types reported were strain and sprain. Kandyan dancers reported the highest number of injuries (p<0.025), contributing to the highest rate of injury due to strains (19.45%). Twirls and prolonged *mandiya* positions are found to be the common mechanisms that cause injuries. Only 10.6% of the participants approached physiotherapy treatments after an injury. According to the findings of the current study, there is a significant rate of dancing-related injuries among Sri Lankan traditional dancers.

## Introduction

Dancing is a challenging form of art that consists of a rhythmically presented succession of purposefully selected series of human movements. Dancing necessitates a lot of practice, balance, motor coordination, and extremes of joint range of motion.

There are various types of traditional dancing styles such as ballet, Bharatanatyam, ballroom, contemporary, hip hop, jazz, tap dance, folk dance, Irish dance, modern dance, and swing dance. A dance form is designed with elements of traditional culture such as music,

**Funding:** The author(s) received no specific funding for this work.

**Competing interests:** The authors have declared that no competing interests exist.

literature, and legends [1]. The origin of traditional dancing in Sri Lanka goes back to immemorial times of native tribes: 'Yakkas', thus, imposing a paramount importance in Sri Lankan fine arts. There are three main styles in Sri Lankan traditional dancing: 'Kandyan dancing' or 'Udarata Natum' of the hill country origin, 'Low country dancing' or 'Pahatharata Natum' of the southern plains origin, and 'Sabaragamuwa dancing' which originated in Sabaragamuwa province [1].

There are some variations in techniques and skills in these three traditional dance styles according to the teachers and different schools (Gurus). Backflips, acrobatics, and leaps are the main skills that are used in the Kandyan dancing style. The famous pirouettes (a twirl on one foot) are the highlights of the performance [2]. Though there are small variations in characteristics and poses of these dancing styles, 'Mandiya' is the most fundamental position of all (Fig 1). This position is achieved by placing the feet three steps away from each other, hips are flexed and abducted, knees are flexed and externally rotated, shoulders are retracted, while the chest is moved forward with arms at the shoulder level. The upper limbs are kept in shoulder abduction, elbow flexion, wrist, and fingers extension. However, the low country dancers bend the trunk more forward and move the feet more outward during the mandiya position, than the other two styles while Sabaragamuwa dancers use greater hip and knee flexion (Fig 1). In addition, Sabaragamuwa dancers practically reach the floor without bending their backs and therefore, Sabaragamuwa dancing style requires extensive training.

Traditional dancers are considered athletes since they perform complex, physically demanding postures, are required to train for extended periods of time and do not have seasonal breaks like for other sports [3]. However, dancers are usually a medically neglected occupational group with a high risk of work-related musculoskeletal disorders (WMSD) [4]. Therefore, the health concerns of traditional dancers are worthy of attention for several reasons because most dancers begin their training at an early age, which poses a significant impact on their future health status [5].

Injury prevention in dancers has become challenging due to lack of research and documentation in the field. Warm-up, cool down and flexibility exercises that are used in traditional dancing as well as approaches followed when returning to dancing after an injury should be carefully studied to minimize the injury rates. Although several studies have been carried out on typical musculoskeletal disorders associated with many traditional dancing forms, no research has been done on dance related injuries in Sri Lanka. Since there are three main types of traditional dancing in Sri Lanka, the injury prevalence, injury mechanism, and the site of injury may vary with regard to the style. Therefore, a cross sectional survey to identify the

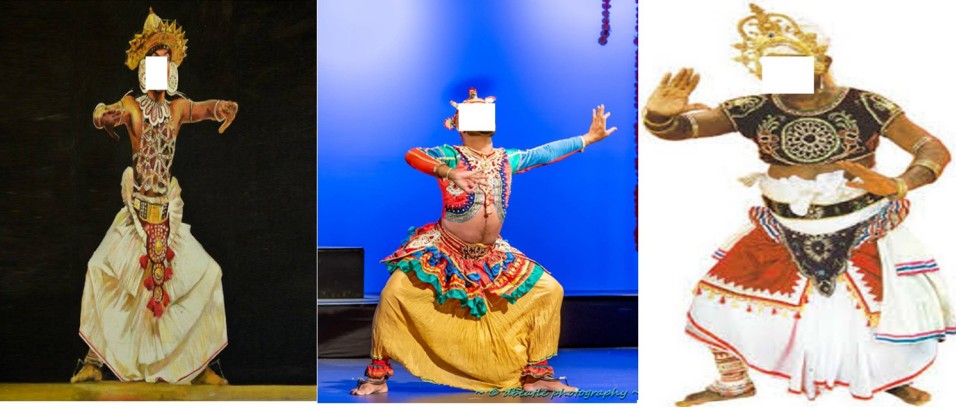

**Fig 1. *Mandiya* position of the three traditional dancing styles in Sri Lanka.** From left to right, Kandyan dancing style, low country Dancing style, and *Sabaragamu* dancing style, respectively.

musculoskeletal injuries prevailing among Sri Lankan traditional dancers was proposed in this study.

Undergraduates who follow Sri Lankan traditional dancing were taken as the study sample in the present study because they show less age-related physiological changes and the young can be considered as healthier since they have minimal health problems. In addition, previous studies have revealed reasons such as dance related injuries, psychological and physiological stress have brought out a negative impact on the quality of life of younger dancers [6]. At present, traditional dancing is taught in a number of dance schools and academies in Sri Lanka with variations in dancing techniques and exercises they follow. In comparison, the university undergraduates follow a similar curriculae and use almost identical dance methods, making them a better population to assess. Thus, this study focused on evaluating the prevalence of common musculoskeletal injuries among university undergraduates following Sri Lankan traditional dancing styles.

## Materials and methods

### Participants

A cross-sectional descriptive study was carried out among the undergraduates of Visual Arts & Design and Performing Arts Unit of the Department of Fine Arts, University of Kelaniya, the Department of Languages, Cultural Studies and Performing Arts, University of Sri Jayawardenapura, the Department of Performing Arts of Sri Palee Campus, University of Colombo and University of the Visual and Performing Arts, Sri Lanka, the four Universities which offer traditional dancing studies in their curriculum.

Both male and female undergraduates between the ages of 18 and 30 who follow one of the three types of Sri Lankan traditional dancing styles as their major subject were recruited. The students who follow other dancing types together with the traditional dancing, those who engage in dancing for less than four years, who do not engage in regular dancing practices, engage in sports activities, have a history of musculoskeletal problems and injuries from sources other than dancing, who have undergone surgeries in the upper and lower limbs, who have deformities and congenital abnormalities in the upper or lower limbs and in the spine were excluded from the study. The sample size for the study was 293 which was calculated using Solvin's formula [7]. The stratified sampling technique was used to select the participants for the study.

### Study procedure

A standard self-administered questionnaire used for assessing musculoskeletal injuries in ballet dancers that included four components related to dancing: socio-demographic data, dance related factors, injury related factors and treatment options was provided to the sample in the form of a Google sheet [3]. From the returned questionnaires, 293 participants who fulfill the inclusion criteria were selected randomly. Informed consent was obtained from all the participants in a written format that was included in the Google sheet. Ethical approval for the study was obtained from the Ethics Review Committee of the Faculty of Allied Health Sciences, University of Peradeniya, Sri Lanka. Also, confidentiality of the participants and the information collected was assured.

### Data analysis

Descriptive data was presented using absolute and relative frequencies of categorical variables such as injury prevalence, type, location and mechanism of injury and data were categorized with relevance to the three traditional dancing types and gender. Continuous variables such as

body mass index (BMI), age, experience in dancing, duration of practice and intervals in one session were presented as Mean ± Standard Deviation. Association between dancing experience, duration of one practice session and the number of intervals in one dancing session with main traditional dancing style and gender of the participants was identified using One-way ANOVA test and t-test while the association between stretching exercises and practicing speed with main traditional dancing style, gender and the academic year of the participants was identified using Chi-square test. Dependency of injury prevalence and related factors on the Sri Lankan traditional dancing styles and gender was also identified using Chi-square test. All statistical analyses were conducted using SPSS statistical software, version 25 (SPSS, IBM Corp. Armonk, NY, USA).

## Results

### Characteristics of the participants

The mean age of the participants was 23 (1.6) years and the majority was females at 64.5%. Among the three types of traditional dancing styles, a majority of 45.1% follows Kandyan dancing. The mean BMI of the participants is 22.34 (3.6) kg/m$^2$ (Table 1).

### Association between dancing-related factors with Sri Lankan traditional dancing styles and the gender of the participants

The results have shown that experience in dancing, the number of hours practiced per day, the number of days practiced per week, intervals taken within one dancing session have a significant association with the dancing style and the gender of the participants (Table 2).

### Prevalence of musculoskeletal injuries in Sri Lankan traditional dancers

In the study sample, 190 (64.8%) dancers reported injuries in dancing. According to the data, strain was the highest reported injury type with 31% prevalence, and the second highest was sprain (17.7%). Subluxation, fractures and other injuries reported were 8.9%, 1.7% and 16.4%, respectively (Fig 2).

### Dependency of injury prevalence and related factors on the Sri Lankan traditional dancing styles and gender

In the study sample, 190 dancers reported injuries in dancing. According to the data, strain was the highest reported injury type with 31.04%, and the second highest was sprain (17.7%).

**Table 1. Characteristics of the participants (n = 293).**

| Variables | No of participants | Percentage (%) |
|---|---|---|
| Age 23 (1.6) years | - | - |
| Gender | | |
| Male | 104 | 35.5 |
| Female | 189 | 64.5 |
| Main dancing style | | |
| Kandyan | 132 | 45.1 |
| Low country | 92 | 31.4 |
| *Sabaragamu* | 69 | 23.5 |
| BMI | | |
| 22.34 (3.6) kg/m$^2$ | - | - |

**Table 2. Association between dancing-related factors with the main traditional dancing style and gender of the participants.**

| Variables | | Main traditional dancing style | | | | | Gender | | |
|---|---|---|---|---|---|---|---|---|---|
| | | Kandyan | Low country | Sabaragamu | *P* value | Effect size | Male | Female | *P* Value |
| | | Mean (SD) | | | | | Mean (SD) | | |
| Dancing experience (years) | | 14.78 (3.66) | 13.34 (3.68) | 12.09 (3.45) | 0.000[a] | | 12.35 (3.49) | 14.44 (3.71) | 0.000[a] |
| Practice hours per day | | 4.06 (2.65) | 5.37 (2.05) | 5.44 (2.03) | 0.000[a] | | 5.16 (2.18) | 4.41 (2.47) | 0.284[a] |
| Practice days per week | | 3.89 (1.52) | 4.71 (1.05) | 4.77 (0.94) | 0.002[a] | | 4.56 (1.17) | 4.24 (1.39) | 0.047[a] |
| Intervals in one dancing session (hours) | | 2.2 (0.62) | 2.02 (0.59) | 1.49 (0.78) | 0.004[a] | | 1.43 (0.76) | 2.2 (0.56) | 0.000[a] |
| Stretching exercise + Warm up exercise | | *n* (%) | | | | | *n* (%) | | |
| | | 103 (35.2) | 65 (22.2) | 49 (16.7) | 0.373[b] | 0.022 | 71 (24.23) | 146 (49.82) | 0.093[b] |
| Stretching exercise + Cool down exercise | | 38 (12.9) | 24 (8.2) | 15 (5.1) | 0.559[b] | 0.032 | 24 (8.19) | 53 (18.08) | 0.355[b] |
| Practicing speed | High | 7 (2.3) | 4 (1.36) | 4 (1.36) | 0.916[b] | 0.053 | 10 (3.41) | 5 (1.7) | 0.013[b] |
| | Moderate | 122 (41.6) | 84 (28.66) | 63 (21.5) | | | 89 (30.37) | 180 (61.43) | |
| | Low | 3 (1.02) | 4 (1.36) | 2 (0.68) | | | 5 (1.7) | 4 (1.36) | |
| Dancing floor | Cement floor | 91 (31.05) | 73 (24.91) | 55 (18.77) | 0.144[b] | 0.008 | 81 (27.64) | 138 (47.09) | 0.665[b] |
| | Tile floor | 33 (11.26) | 19 (6.48) | 13 (4.43) | | | 21 (7.16) | 44 (15.01) | |
| | Sand floor | 4 (1.36) | 0 (0) | 0 (0) | | | 2 (0.68) | 2 (0.68) | |

Data shown as Mean (SD) or *n* (%) with 95% confidence interval. [a]One-way ANOVA. [b]Chi-square test

Subluxation, fractures and other injuries reported were 8.9%, 1.7% and 16.4%, respectively. The highest rate of injuries by strain (19.45%) was reported by students who follow Kandyan Dancing whereas; sprain was more common among Low country and Sabaragamu dancers.

When the prevalence of injuries in each dancing style was assessed, strains, subluxations, sprains, and other injuries showed significant associations with the type of traditional dancing style. Males reported a substantially higher overall injury rate than that of females according to the prevalence of injuries with relevance to the gender. Further, sprain is the most common type of injury in males, while strain is the most common in females (Table 3).

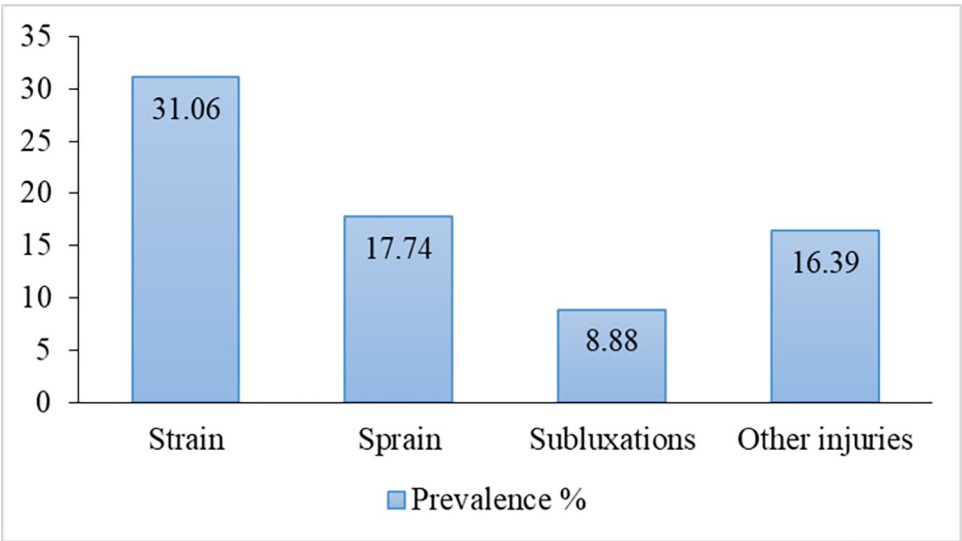

**Fig 2. Prevalence of musculoskeletal injuries in Sri Lankan traditional dancers.**

**Table 3. Association between injury prevalence and related factors with the main traditional dancing style and the gender of the participants.**

| Variables | Main traditional dancing style | | | | | Gender | | |
|---|---|---|---|---|---|---|---|---|
| | Kandyan | Low country | Sabaragamu | P–value | Effect size | Male | Female | P -value |
| Injuries during dancing n = 190 n (%) | 96 (32.76%) | 51 (17.40%) | 43 (14.67%) | 0.025[b] | 0.025 | 82 (27.98%) | 108 (36.86%) | 0.000[b] |
| Type of injuries n (%) | | | | | | | | |
| Fracture n = 5 | 1 (0.34%) | 2 (0.68%) | 2 (0.68%) | 0.493[b] | 0.02 | 3 (1.02%) | 2 (0.68%) | 0.248[b] |
| Strain n = 91 | 57 (19.45%) | 21 (7.16%) | 13 (4.43%) | 0.000[b] | 0 | 36 (12.28%) | 55 (18.77%) | 0.329[b] |
| Subluxation n = 26 | 7 (2.38%) | 8 (2.73%) | 11 (3.75%) | 0.042[b] | 0.002 | 17 (5.8%) | 9 (3.07%) | 0.001[b] |
| Sprain n = 52 | 15 (5.11%) | 19 (6.48%) | 18 6.14%) | 0.023[b] | 0.001 | 32 (10.92%) | 20 (6.82%) | 0.000[b] |
| Other injuries n = 48 | 30 (10.23%) | 10 (3.41%) | 8 (2.73%) | 0.029[b] | 0.001 | 18 (6.14%) | 30 (10.25%) | 0.751[b] |

Data shown as n (%). [b]Chi-square test

## Association between the mechanism of injury, location of the injury and the Sri Lankan traditional dancing style and the gender

Twirls and prolonged *mandiya* positions were the common mechanisms that cause a higher injury rate among dancers. Nevertheless, prolonged *mandiya* position, jumps and other combined movements were found to have a potent correlation with the higher injury rate reported among Kandyan dancers while backflips, acrobatics, and twirls were the movements causing more injuries among Low country dancers. Prolonged *mandiya* position, backflips and acrobatics cause significantly higher injury rates in males compared to that in females. Females reported a higher injury rate in neck, shoulder, upper chest, elbow, hand, thigh, knee, and leg than males. However, ankle, knee, and thigh are the most common injury sites reported by the traditional dancers regardless of the type of dancing style and gender. Further, neck, lumbar spine, thigh, knee and ankle were the most prevalent injury sites among Kandyan dancers (Table 4).

## Dancers' approach to treatment options following an injury

Only 16.3% of the participants had received Physiotherapy treatments after an injury, while 63.15% opted for other types of care.

## Discussion

Traditional forms of dancing in Sri Lanka are high-performance physical activities that require an advanced level of technical skills. These dancers frequently place great stress on tendons, muscles, bones, and joints. Therefore, they are a high-risk group for chronic injuries. Though there are studies explaining common musculoskeletal problems associated with various traditional dancing types, no research has been carried out in Sri Lanka assessing the traditional dancers. Thus, this study focused on assessing the prevalence of musculoskeletal injuries among university undergraduates who follow Sri Lankan traditional dancing.

According to the study results, the majority of dancers practiced warm-up exercises and only 20% of dancers followed cool-down exercises after a dancing session. Although the importance of warm-up and cool-down in general exercise and sports is widely acknowledged, awareness of their importance among Sri Lankan traditional dancers is poor. Even though the majority of dancers regularly performed the warm-up exercises, musculoskeletal pain was widely reported among them which highlights the fact that musculotendinous injury does not purely result from lack of elasticity but when tension demands of the muscle exceeds the tension generating capability of the muscle [8].

**Table 4. Association between the mechanism of injury and location of injury with the main traditional dancing style and the gender of the participants.**

| Variables | Main traditional dancing style | | | | | Gender | | |
|---|---|---|---|---|---|---|---|---|
| | Kandyan | Low country | Sabaragamu | *P*–value | Effect size | Male | Female | *P* value |
| | Mechanism of injury *n* (%) | | | | | | | |
| Prolong mandiya position n = 43 | 20 (6.8) | 5 (1.7) | 18 (6.1) | 0.001[b] | 0 | 28 (9.55) | 15 (5.1) | 0.000[b] |
| Back flips n = 25 | 7 (2.3) | 11 (3.75) | 7(2.3) | 0.185[b] | 0.01 | 20 (6.82) | 5 (1.7) | 0.000[b] |
| Acrobatics n = 25 | 7 (2.3) | 11 (3.75) | 7 (2.3) | 0.185[b] | 0.01 | 20 6.82) | 5 (1.7) | 0.000[b] |
| Leaps n = 24 | 8 (2.7) | 8 (2.7) | 8 (2.7) | 0.389[b] | 0.022 | 10 (3.41) | 14 (4.7) | 0.510[b] |
| Twirls n = 46 | 14 (4.77) | 23 (7.8) | 9 (3.07) | 0.011[b] | 0.0006 | 20 (6.82) | 26 (8.87) | 0.218[b] |
| Jumps n = 29 | 24 (8.19) | 2 (0.68) | 3 (1.02) | 0.000[b] | 0 | 7 (2.38) | 22 (7.5) | 0.178[b] |
| Other mechanisms n = 72 | 49 (16.7) | 16 (5.4) | 7 (2.3) | 0.000[b] | 0 | 22 (7.5) | 50 (17.06) | 0.313[b] |
| | Location of injury *n* (%) | | | | | | | |
| Face n = 0 | 0 | 0 | 0 | .[b] | - | 0 | 0 | .[b] |
| Neck n = 14 | 12 (4.09) | 2 (0.68) | 0 | 0.006[b] | 0.0003 | 4 (1.36) | 10 (3.41) | 0.579[b] |
| Shoulder and upper chest n = 21 | 9 (3.07) | 10 (3.4) | 2 (0.68) | 0.149[b] | 0.008 | 10 (3.41) | 11 (3.75) | 0.228[b] |
| Elbow n = 11 | 6 (2.04) | 4 (1.36) | 1 (0.34) | 0.514[b] | 0.030 | 3 (1.03) | 8 (2.73) | 0.561[b] |
| Forearm n = 4 | 2 (0.68) | 1 (0.34) | 1 (0.34) | 0.961[b] | 0.056 | 2 (0.68) | 2 (0.68) | 0.542[b] |
| Hand n = 5 | 2 (0.68) | 3 (1.02) | 0(0) | 0.279[b] | 0.016 | 0 | 5 (1.7) | 0.094[b] |
| Lumbar n = 36 | 20 (6.8) | 9 (3.07) | 7 (2.3) | 0.400[b] | 0.023 | 18 (6.14) | 18 (6.14) | 0.052[b] |
| Pelvic n = 0 | 0 | 0 | 0 | .[b] | - | 0 | 0 | .[b] |
| Buttocks n = 12 | 8 (2.7) | 2 (0.68) | 2 (0.68) | 0.299[b] | 0.017 | 7 (2.38) | 5 (1.7) | 0.091[b] |
| Thigh n = 51 | 30 (10.23) | 9 (3.07) | 12 (4.09) | 0.042[b] | 0.002 | 25 (8.53) | 26 (8.87) | 0.026[b] |
| Knee n = 52 | 29 (9.89) | 12 (4.09) | 11 (3.75) | 0.206[b] | 0.012 | 19 (6.48) | 33 (11.2) | 0.862[b] |
| Leg n = 24 | 15 (5.11) | 7 (2.3) | 2 (0.68) | 0.112[b] | 0.0065 | 8 (2.73) | 16 (5.46) | 0.817[b] |
| Ankle n = 55 | 21 (7.16) | 19 (6.48) | 15 (5.11) | 0.517[b] | 0.03 | 28 (9.55) | 27 (9.21) | 0.008[b] |
| Foot n = 0 | 0 | 0 | 0 | .[b] | - | 0 | 0 | .[b] |
| Abdomen n = 0 | 0 | 0 | 0 | .[b] | - | 0 | 0 | .[b] |

Data shown *n* (%). [b]Chi-square test

In the current study, dance-related injuries were reported by 64.8% of the participants out of which strain and sprain injuries were the most common injuries while fractures were less common. This could be due to the fact that all three styles of dancing employ extreme motions that can injure ligaments and muscles. Despite dancing being an athletic exercise, fractures are uncommon among dancers as they maintain a lower contact force [9]. Soft tissues, on the other hand, are subjected to prolonged stress during dancing which may be a reason for the higher rate of sprains and strains among dancers [10]. Similarly, a study conducted in Korea reported that the prevalence of strains and sprains was as high as 89.6% among breakdancers [11].

In this study, back, knee and ankle were the most frequently reported sites of injuries. These findings are in line with recent studies conducted in India to assess dance related injuries in Barathanatyam dancing style [5, 12]. Similarly, a systematic review on epidemiology on musculoskeletal injuries in Indian classical dancers revealed a higher incidence of back injuries followed by injuries to the knee and ankle in all forms of classical dancing [13]. Apart from Indian classical dancing, dancing styles such as Irish, Flamenco, Turkish, and Morris dance from Great Britain reported a higher incidence of lower limb injuries [14]. In addition, the current study found that injuries in neck, thigh, knee, and ankle are prevalent among Kandyan dancers with a significantly higher percentages of neck and thigh injuries. Further, excessive head movements cause strain on neck muscles due to extreme movements of spine when

arching the head or neck [15]. Furthermore, it was found that ankle is the most common injury site among Low country and *Sabaragamu* dancers. Other studies show that Ballet dancers also found to have more injuries in the spine, shoulder, ankle, and foot [3, 16]. Likewise, modern dancers reported the same areas as the sites which are more prone for injuries including lower back and foot [17, 18]. The ankle has been identified as a common site for injuries in many of these studies as the body weight shifts to the ground via the ankle joint, and ankle is subjected to sudden torsions while dancing [19, 20]. Stretching exercises and active range of motion exercises through the entire joint play should be encouraged before dance sessions to enhance the flexibility of the muscles around the ankle and other areas reported to be more susceptible to injuries. However, in dance styles like Pole dance and Belly dance, the most common injury sites are the shoulder, wrist and pelvic area, respectively [14, 21]. Although the lower limb is the most prevalent injury site in most dance genres, it varies depending on the dancing movements used in each dancing style.

In the current study, it was evident that as vigorous, high-speed movements like twirls and prolonged *mandiya* position are more frequently used in Kandyan form of dancing where the injury rate is higher compared to Low country and *Sabaragamu* dancers. Low country dancers, on the other hand, sustain fewer injuries than the other two categories of dancers owing to the slower pace movements and performing more warm-up exercises than *Sabaragamuwa* and Kandyan dancers. A report published on the importance of warm-up and cool down exercises as well verified that the risk of injuries while dancing were lowered with effective warm-up exercises [22].

In traditional Sri Lankan styles, only the highly experienced dancers use mechanisms like jumping, back flips, acrobatics and leaps while majority use *mandiya* position and twirls [5]. According to the results of the study, twirls and prolonged *mandiya* position were found to be the common movements that cause higher rate of injuries among traditional dancers in Sri Lanka. Twirls caused majority of injuries in low country dancers as it is a basic movement of the style. On the contrary, *Sabaragamuwa* dancers suffer from injuries mostly as a result of prolonged *mandiya* position. Likewise, among Indian Baratantyam dancer "*Aramandi*" position which is similar to the prolonged *mandiya* position in Sri Lankan traditional dancing, has caused most of the injuries among Indian Baratanatyam dancers [5]. Prolonged movements overload the knee and ankle joints and results in micro traumas to the ligaments and muscles that surrounds them [3]. The current study clearly shows that holding of prolonged positions during dancing cause greater rate of injuries in Sri Lankan traditional dancing, with the knee and ankle joints being the most vulnerable injury sites. In addition, both the literature and the current study show that repetitive and vigorous movements results in injuries [3]. However, the mechanism of dance related injuries is different among Ballet dancers where jumps, lifting and pointe work are the common movement patterns [23]. Hence, previous studies and the current research suggest the mechanism of injury varies depending on the dancing style.

The present study revealed that males experience a higher rate of injuries during dancing which might have arisen from factors such as lack of experience than females, using more vigorous movements, performing fewer warm-up and cool-down exercises, lengthy practice sessions per day than females, having brief breaks between two dancing sessions, and practicing at a faster pace than female dancers. These findings are compatible with the studies conducted on Ballet dancers and Ballroom dancers that found high injury rate among the male dancers [23, 24].

According to the literature, dancers who dance on hard flooring are more likely to get injuried [9, 25]. A biomechanical study comparing three different surfaces used by Indian Bharathnatyam dancers found that non reliance surfaces like concrete produce higher impact forces while dancing and it was mentioned that the type of the dancing floor is an important factor to

be considered in order to minimize dance related injuries [26]. The current study also found that dancers who perform on cement floors are more prone for injuries.

Physiotherapy treatment is the optimum approach for musculoskeletal injuries. However, according to the results of the current study, only 16.3% of the injured dancers had undergone physiotherapy treatments. In addition, according to studies that reported dance related injuries dancers felt compulsive to return to dance before their injuries were healed as they had invested considerable amount of time in dancing [5]. Further, when the injured ligament or muscle does not receive adequate rest and appropriate exercise there is a frequent recurrence of an injury. Thus, dancing while having biomechanical dysfunction or muscle imbalance can lead to severe consequences that require the cessation of dance activities, which in turn increase the stress levels of dancers [5, 27].

The present study had certain limitations. First, due to the COVID-19 pandemic, data was collected via a Google sheet which might have hindered the true responses of the participants. Further, for the past two years all of the study participants have had less frequent chances to perform and practice dancing. Therefore, it may have been difficult for them to recall their memory on dancing related injuries.

## Conclusion

According to the findings of the current study, there is a significant rate of dancing-related injuries among Sri Lankan traditional dancers. Female dancers showed a higher rate of injuries than male dancers; while Kandyan dancers had the highest injury rate. Most of the dancers have suffered from soft tissue injuries like strains and sprains that are prevalent in other dancing styles in the world. The thigh and knee are the most common injury sites among Sri Lankan traditional dancers. In addition, this study has identified that only a few dancers approached physiotherapy treatments following an injury. Moreover this study paved the way for a number of further studies aimed at reducing dance-related injuries and enhancing the career and quality of life of Sri Lankan traditional dancers. The identification of various dancing-related factors that cause injuries in Sri Lankan traditional dancing, such as the type of common injuries, most vulnerable sites for injuries, and the movement patterns that cause higher injury rates, may help to improve therapeutic actions aimed at the proper biomechanics and improved performance.

## Supporting information

**S1 File. Questionnaire.**
(DOCX)

## Acknowledgments

We thank Prabodani Samarakoon, Instructor in English, University of Peradeniya, Sri Lanka for her assistance in proof reading and language editing.

## Author Contributions

**Conceptualization:** Geethika Chathurani, Yasantha B. Dassanayake, Lahiru S. Gunarathna.

**Data curation:** Geethika Chathurani, Yasantha B. Dassanayake, Sanduni N. Fernando, Lahiru S. Gunarathna, Lakshani K. Gunarathne, Dilhari Senarath, Surangika I. Wadugodapitiya.

**Formal analysis:** Geethika Chathurani, Yasantha B. Dassanayake, Sanduni N. Fernando, Lahiru S. Gunarathna, Lakshani K. Gunarathne, Nadheera C. Chandrasekara.

**Investigation:** Sanduni N. Fernando, Lahiru S. Gunarathna, Lakshani K. Gunarathne, Nadheera C. Chandrasekara.

**Methodology:** Geethika Chathurani, Yasantha B. Dassanayake, Sanduni N. Fernando, Lahiru S. Gunarathna, Lakshani K. Gunarathne, Nadheera C. Chandrasekara, Dilhari Senarath, Surangika I. Wadugodapitiya.

**Project administration:** Surangika I. Wadugodapitiya.

**Supervision:** Dilhari Senarath, Surangika I. Wadugodapitiya.

**Writing – original draft:** Geethika Chathurani, Yasantha B. Dassanayake, Sanduni N. Fernando, Lahiru S. Gunarathna, Lakshani K. Gunarathne, Nadheera C. Chandrasekara.

**Writing – review & editing:** Dilhari Senarath, Surangika I. Wadugodapitiya.

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
