## [Decision Letter · Decision Letter 0]

25 Jan 2023

PONE-D-22-35570Prevalence of musculoskeletal injuries among university undergraduates following Sri Lankan traditional dancingPLOS ONE

Dear Dr. Wadugodapitiya,

Thank you for submitting your manuscript to PLOS ONE. After careful consideration, we feel that it has merit but does not fully meet PLOS ONE’s publication criteria as it currently stands. Therefore, we invite you to submit a revised version of the manuscript that addresses the points raised during the review process. Some of the comments provided will require significant revisions of the manuscript.

We look forward to receiving your revised manuscript.

Kind regards,

Andrea Dell'Isola

Academic Editor

PLOS ONE

Journal Requirements:

2. We note that Figure 1 includes an image of a participant in the study. 

As per the PLOS ONE policy (http://journals.plos.org/plosone/s/submission-guidelines#loc-human-subjects-research) on papers that include identifying, or potentially identifying, information, the individual(s) or parent(s)/guardian(s) must be informed of the terms of the PLOS open-access (CC-BY) license and provide specific permission for publication of these details under the terms of this license. Please download the Consent Form for Publication in a PLOS Journal (http://journals.plos.org/plosone/s/file?id=8ce6/plos-consent-form-english.pdf). The signed consent form should not be submitted with the manuscript, but should be securely filed in the individual's case notes. 

Please amend the methods section and ethics statement of the manuscript to explicitly state that the participant has provided consent for publication: “The individual in this manuscript has given written informed consent (as outlined in PLOS consent form) to publish these case details”. 

**Additional Editor Comments:**

The study focuses on “evaluating the prevalence of common musculoskeletal injuries among university undergraduates following Sri Lankan traditional dancing styles”. However, a large part of the results and discussions are dedicated to analysing the association between other factors and the injury. Please revise the manuscript so that the aims are correctly aligned with the analysis, results and discussions. Also please keep a consistent order in presenting and discussing the results (the other should be based on the aims).

Currently, several aspects of the methodology are unclear and need to be specified. Please restructure the manuscript to follow guidelines for reporting e-surveys (e.g. Eysenbach G. Improving the quality of Web surveys: the Checklist for Reporting Results of Internet E-Surveys (CHERRIES). J Med Internet Res. 2004 Sep 29;6(3):e34. doi: 10.2196/jmir.6.3.e34. Erratum in: doi:10.2196/jmir.2042. PMID: 15471760; PMCID: PMC1550605.) and provide a copy of the checklist (https://www.elsevier.com/__data/promis_misc/JMIG_CHERRIES.docx)

Can the authors clarify how they accounted for multiple testing in the analysis?

If the main focus of the study is the prevalence of injuries, I would suggest estimating prevalence rather than relying on the chi-square test to compare the distribution between groups.

Please report measures of uncertainty for your estimates, possibly 95%CI as they can be easily interpreted by most of the readers.

Please carefully revise the manuscript to make sure that the methods, results, and discussions are aligned. Also, make sure that all the statements correctly reflect the results (see comments from reviewer 2)

Please specify in your data availability statement where the data are stored.

Reviewers' comments:

Reviewer's Responses to Questions

**Comments to the Author**

1. Is the manuscript technically sound, and do the data support the conclusions?

Reviewer #1: Yes

Reviewer #2: Yes

2. Has the statistical analysis been performed appropriately and rigorously? 

Reviewer #1: Yes

Reviewer #2: No

3. Have the authors made all data underlying the findings in their manuscript fully available?

Reviewer #1: Yes

Reviewer #2: Yes

4. Is the manuscript presented in an intelligible fashion and written in standard English?

Reviewer #1: Yes

Reviewer #2: Yes

5. Review Comments to the Author

Reviewer #1: Manuscript ID: PONE-D-22-35570

Manuscript title: Prevalence of Musculoskeletal Injuries among University Undergraduates Following Sri Lankan Traditional Dancing

Comments

This manuscript reports a study designed to evaluate the prevalence of common musculoskeletal injuries among university undergraduates following Sri Lankan traditional dancing styles. The manuscript is generally well-written and is of interest to the field. I have only a few minor suggestions for the authors to consider.

Minor comments

1. Abstract. This section is usually a single paragraph, please double-check the journal’s instructions for authors.

2. Tables 2 to 4. Very low p-values should be reported as <0.001 (rather than 0.000).

3. When reporting summary statistics of categorical data, please report consistently in the format “n (%)”.

Reviewer #2: The study assessed musculoskeletal injuries prevalence among undergraduate dancers. Dancing is an important occupation prone to injuries. Therefore, the study is relevant. However, the statistical analysis and results need overhauling.

Abstract: Can you make the abstract a paragraph?

Introduction

Merge paragraphs 1 and 2 of introduction.

Methods

1. What is the validity of this questionnaire? Please report the psychometric properties of the questionnaire.

2. "From the returned questionnaires, 293 participants who fulfill the inclusion criteria were selected randomly". You stated that you used stratify sampling to select 293 participants? This statement suggests that more participants were included out of which 293 participants were randomly selected. Can you clarify?

3. Statistical analysis: ANOVA test assesses the mean difference in three or more groups and not association. please reconstruct the statement in the analysis section.

Results

Report of tables 2 and 3 need review. The caption of table 2 needs review.

Discussion

Can you start the discussion with a summary of your findings? you may delete paragraph one.

See the attached for other comments.

6. PLOS authors have the option to publish the peer review history of their article (what does this mean?). If published, this will include your full peer review and any attached files.

Reviewer #1: **Yes: **Arthur de Sá Ferreira

Reviewer #2: **Yes: **Dr. Oyewole O. Olufemi

---

## [Author Response · Author response to Decision Letter 0]

8 Jun 2023

Responses to Reviewers 

Thank you for giving an opportunity to submit the revisions.

Editor’s comments

Journal Requirements:

The manuscript is now revised according to the guidelines of the journal

2. We note that Figure 1 includes an image of a participant in the study

In order to prevent identification, the individuals' whole faces have been covered in the fig 1.

Also, written informed consent was obtained from the individuals (as outlined in PLOS consent form) in the Figure 1 of this manuscript to publish their photos. 

It is now mentioned in the subheading “study procedure” under the section “Materials and methods”

Additional Comments:

1. This study focuses on “evaluating the prevalence of common musculoskeletal injuries among university undergraduates following Sri Lankan traditional dancing styles”. However, a large part of the results and discussions are dedicated to analysing the association between other factors and the injury. Please revise the manuscript so that the aims are correctly aligned with the analysis, results and discussions. Also please keep a consistent order in presenting and discussing the results (the other should be based on the aims).

The tittle of the manuscript and the aim are now edited.

The manuscript is now revised accordingly (analysis, results and discussion). Also, a sub heading (Prevalence of musculoskeletal injuries in Sri Lankan traditional dancers) is added, to report the injury prevalence with a table (Table 3).

2. Currently, several aspects of the methodology are unclear and need to be specified. Please restructure the manuscript to follow guidelines for reporting e-surveys (e.g. Eysenbach G. Improving the quality of Web surveys: the Checklist for Reporting Results of Internet E-Surveys (CHERRIES). J Med Internet Res. 2004 Sep 29;6(3):e34. doi: 10.2196/jmir.6.3.e34. Erratum in: doi:10.2196/jmir.2042. PMID: 15471760; PMCID: PMC1550605.) and provide a copy of the checklist (https://www.elsevier.com/__data/promis_misc/JMIG_CHERRIES.docx)

The check list is now filled and changed the materials and methods section accordingly (The check list is attached to the response to reviewer letter in the uploaded documents list as Annexure A)

3. Can the authors clarify how they accounted for multiple testing in the analysis?

Since there are three types in Sri Lankan traditional dancing: Kandyan, low country and Sabaragamuwa, to compare the factors between the groups multiple testing like ANOVA was used

4. If the main focus of the study is the prevalence of injuries, I would suggest estimating prevalence rather than relying on the chi-square test to compare the distribution between groups.

Injury prevalence is now added separately in a subheading. Injury prevalence based on the traditional dancing type and the gender has compared separately

5. Please report measures of uncertainty for your estimates, possibly 95%CI as they can be easily interpreted by most of the readers.

This statement is now added to the data analysis section under materials and methods section

6. Please carefully revise the manuscript to make sure that the methods, results, and discussions are aligned. Also, make sure that all the statements correctly reflect the results (see comments from reviewer 2)

Revised the methods, results and discussion sections accordingly 

Please specify in your data availability statement where the data are stored.

Included a statement regarding the data storing in the study procedure section under materials and methods

Reviewer 1 

Minor comments 

Abstract: This section is usually a single paragraph, please double-check the journal’s instructions for authors.

Revised the abstract accordingly

Tables 2 to 4: Very low p-values should be reported as <0.001 (rather than 0.000)

Corrections made in the tables.

When reporting summary statistics of categorical data, please report consistently in the format “n (%)”.

Corrections made

Reviewer 2

Abstract: 

Can you make the abstract a paragraph?

Revised the abstract

Introduction:

Merge paragraphs 1 and 2 of introduction

Correction made

Methods

i. What is the validity of this questionnaire? Please report the psychometric properties of the questionnaire

With the consent of the original authors, a validated self-administered questionnaire used in a study evaluating musculoskeletal injuries in ballet dancers was provided to the study population in this study 

ii. "From the returned questionnaires, 293 participants who fulfill the inclusion criteria were selected randomly". You stated that you used stratify sampling to select 293 participants? This statement suggests that more participants were included out of which 293 participants were randomly selected. Can you clarify?

This statement is now corrected under the section “participants” in the materials and methods. 

iii. Statistical analysis: ANOVA test assesses the mean difference in three or more groups and not association. please reconstruct the statement in the analysis section.

This is now corrected under the section “data analysis” in the materials and methods 

iv. Results

Report of tables 2 and 3 need review. The caption of table 2 needs review

The report of the Tables 2 and 3 and the captions are now reviewed 

v. Discussion

Can you start the discussion with a summary of your findings? you may delete paragraph one.

Corrections made accordingly

---

## [Decision Letter · Decision Letter 1]

20 Jun 2023

Prevalence and associated factors of musculoskeletal injuries among university undergraduates following Sri Lankan traditional dancing

PONE-D-22-35570R1

Dear Dr. Wadugodapitiya,

We’re pleased to inform you that your manuscript has been judged scientifically suitable for publication and will be formally accepted for publication once the comments from reviewer two will be addressed and all outstanding technical requirements will be met.

Kind regards,

Andrea Dell'Isola

Academic Editor

PLOS ONE

Reviewers' comments:

Reviewer's Responses to Questions

**Comments to the Author**

1. If the authors have adequately addressed your comments raised in a previous round of review and you feel that this manuscript is now acceptable for publication, you may indicate that here to bypass the “Comments to the Author” section, enter your conflict of interest statement in the “Confidential to Editor” section, and submit your "Accept" recommendation.

Reviewer #1: All comments have been addressed

Reviewer #2: (No Response)

2. Is the manuscript technically sound, and do the data support the conclusions?

Reviewer #1: Yes

Reviewer #2: Yes

3. Has the statistical analysis been performed appropriately and rigorously? 

Reviewer #1: Yes

Reviewer #2: Yes

4. Have the authors made all data underlying the findings in their manuscript fully available?

Reviewer #1: Yes

Reviewer #2: Yes

5. Is the manuscript presented in an intelligible fashion and written in standard English?

Reviewer #1: Yes

Reviewer #2: Yes

6. Review Comments to the Author

Reviewer #1: (No Response)

Reviewer #2: The manuscript has improved. Minor corrections needed.

Abstract

i. "Among the three types of traditional dancing styles, the majority were following Kandyan dancing: 45.1%" (lines 32-33). Please change 'majority' to 'many'.

ii. "Out of the study sample, 190 dancers reported injuries with females indicating the highest rate of injuries" (lines 33-34). Please insert the percentage (%) of 190.

Data analysis

".....................using One-way ANOVA test....." (lines 148-148). change to "...........using One-way ANOVA and t-test..."

Results

i. Table 3 is better presented in figure.

ii. No 95% CI was indicated in Tables 4 & 5.

iii. One-way ANOVA was not indicated in Table 5. Delete the footnote, it is not applicable here.

Conclusion

i. "Female dancers showed a higher rate of injuries than female dancers" (lines 316-317). change to "Female dancers showed a higher rate of injuries than male dancers".

7. PLOS authors have the option to publish the peer review history of their article (what does this mean?). If published, this will include your full peer review and any attached files.

Reviewer #1: **Yes: **Arthur de Sá Ferreira

Reviewer #2: No

---

## [Editor Report · Acceptance letter]

2 Aug 2023

PONE-D-22-35570R1 

Prevalence of Musculoskeletal Injuries among University Undergraduates Following Sri Lankan Traditional Dancing 

Dear Dr. Wadugodapitiya:

I'm pleased to inform you that your manuscript has been deemed suitable for publication in PLOS ONE. Congratulations! Your manuscript is now with our production department. 

Kind regards, 

on behalf of

Assoc Prof Andrea Dell'Isola 

Academic Editor

PLOS ONE